



# Estimation of the atmospheric hydroxyl radical oxidative capacity using multiple hydrofluorocarbons (HFCs)

Rona L. Thompson[1], Stephen A. Montzka[2], Martin K. Vollmer[3], Jgor Arduini[4], Molly Crotwell[2,5], Paul Krummel[6], Chris Lunder[1], Jens Mühle[7], Simon O'Doherty[8], Ronald G. Prinn[9], Stefan Reimann[3], Isaac Vimont[2], Hsiang Wang[10], Ray F. Weiss[7], and Dickon Young[8]

[1]NILU - Norsk Institutt for Luftforskning, Kjeller, 2007, Norway
[2]NOAA, Earth System Research Laboratory, Boulder, CO, USA
[3]Empa, Swiss Federal Laboratories for Materials Science and Technologies, Switzerland
[4]Università degli Studi di Urbino, Chemical Sciences Section, Urbino, Italy
[5]Cooperative Institute for Research in Environmental Sciences, University of Colorado, Boulder, CO USA
[6]CSIRO Environment, Aspendale, Victoria, Australia
[7]Scripps Institution of Oceanography, University of California San Diego, La Jolla, CA, USA
[8]School of Chemistry, University of Bristol, UK
[9]Center for Global Change Science, Massachusetts Institute of Technology, Cambridge, MA, USA
[10]School of Earth and Atmospheric Sciences, Georgia Institute of Technology, GA, USA

*Correspondence to*: Rona L. Thompson (rlt@nilu.no)

**Abstract.** The hydroxyl radical (OH) largely determines the atmosphere's oxidative capacity and, thus, the lifetimes of numerous trace gases, including methane ($CH_4$). Hitherto, observation-based approaches for estimating the atmospheric oxidative capacity have primarily relied on using methyl chloroform (MCF), but as the atmospheric abundance of MCF has declined, the uncertainties associated with this method have increased. In this study, we examine the use of five hydrofluorocarbons (HFCs) (HFC-134a, HFC-152a, HFC-365mfc, HFC-245fa and HFC-32) in multi-species inversions, which assimilate three HFCs simultaneously, as an alternative method to estimate atmospheric OH. We find robust estimates of OH regardless of which combination of three HFCs are used in the inversions. Our results show that OH has remained fairly stable during our study period from 2004 to 2021, with variations of <2% and no significant trend. Inversions including HFC-32 and HFC-152a (the shortest-lived species) indicate a small reduction in OH in 2020 ($1.6\% \pm 0.9\%$ relative to the mean over 2004-2021 and $0.6 \pm 0.9\%$ lower than in 2019), but considering all inversions, the reduction was only $0.5 \pm 1.1\%$ and OH was at a similar level to that in 2019.

## 1 Introduction

The oxidative capacity of the atmosphere is largely controlled by the concentration of hydroxyl radicals (OH) in the troposphere (Lelieveld et al., 2016) and determines the atmospheric lifetime of numerous species including methane ($CH_4$), other volatile organic compounds and carbon monoxide (CO). Since the atmospheric lifetime of OH is very short, ~1 second (Lelieveld et al., 2004), it is impossible to directly determine the concentration of OH on temporal and spatial scales relevant



for understanding the overall tropospheric oxidation capacity and its integrated effect on longer-lived species, such as CH$_4$ or CO. Therefore, OH concentrations are estimated either by atmospheric chemistry transport models (ACTMs) e.g. (Anderson et al., 2021; Nicely et al., 2017; Lelieveld et al., 2016; He et al., 2021; Voulgarakis et al., 2013), or via proxy methods, which infer the OH oxidative capacity from changes in a reactant species, most commonly methyl chloroform (MCF) (e.g. Montzka et al., 2011; Naus et al., 2021; Bousquet et al., 2005; Krol and Lelieveld, 2003; Rigby et al., 2017; McNorton et al., 2016;

Prinn et al., 2001, 1992; Patra et al., 2021).

ACTMs estimate OH concentrations by modelling its chemical formation and loss in the atmosphere. There are both primary and secondary formation mechanisms: the primary mechanism is via photolysis of ozone (O$_3$) and the secondary is via recycling of OH in radical reaction chains and accounts for about two thirds of the total OH source (Lelieveld et al., 2016).

There are, however, significant differences in OH abundance and distribution between models, which has been attributed to differences in chemical mechanisms, frequency of O$_3$ photolysis, abundances of O$_3$ and CO (Nicely et al., 2017), humidity, and NOx concentrations (Wild et al., 2020).

For many years, proxy methods have inferred the OH oxidative capacity using observations of MCF partly because relatively

accurate emission magnitudes were derivable from its reported production. The suitability of MCF as tracer for examining changes in OH was enhanced after 1998 because MCF has nearly negligible emissions after that time so that its atmospheric abundance has been dominated by its reaction with OH (Montzka et al., 2011; Naus et al., 2021). However, with the decreasing atmospheric abundance it is possible that the ocean has started outgassing MCF, and there are potentially ongoing low-level emissions from continued and exempted industrial production of MCF, both of which lead to considerable uncertainties in the

MCF source (Patra et al., 2021; Wennberg et al., 2004; Naus et al., 2021). Moreover, with the decreasing atmospheric abundance of MCF (~1.35 ppt in 2021), the variability in the measurements means that the relative uncertainty in the global abundance is also becoming more important. This can be seen from the expression for the oxidative capacity:

$$k(t)[OH] = \frac{E}{B} - \frac{\frac{dB}{dt}}{B} \qquad (1)$$

where $E$ is the emission and $B$ is the abundance of MCF. Considering the uncertainty in the emissions, and the continuously

declining MCF abundance, the uncertainty in OH derived from this method will become more problematic in the future.

The considerable uncertainty in both derivations of OH, from ACTMs and via MCF, can be seen in the disparate estimates of OH variability and trends. Most estimates based on MCF are quite sensitive to the assumed or derived emission trends over time, and some indicate a decrease in OH from about 2005 (Rigby et al., 2017; McNorton et al., 2016) while one study shows

no trend (Patra et al., 2021), whereas most estimates from ACTMs indicate an increasing trend in OH (Stevenson et al., 2020; Zhao et al., 2020; Dalsøren et al., 2016). Having a better constraint on OH is of upmost importance in order to better understand





the recent changes in CH$_4$, which since 2007 has been rising in the atmosphere and at an increasing rate, with the largest recorded annual increases in 2020, 2021 and 2022 (Peng et al., 2022)(https://gml.noaa.gov/ccgg/trends_ch4/).

In this study, we explore the possibility of using multiple hydrofluorocarbon (HFC) species simultaneously to constrain the OH oxidative capacity. Some individual HFC and HCFC alternatives to MCF have been explored in the past (Huang and Prinn, 2002), and a multi-species approach has also been suggested by Liang et al. (2017) as an alternative to MCF but has not yet been implemented. HFCs are synthetic species used in a variety of applications and were developed as substitutes for ozone-depleting chlorofluorocarbons (CFCs) and hydrochlorofluorocarbons HCFCs (Campbell et al., 2005). We selected HFCs that

have an atmospheric sink primarily due to OH (i.e. >99% of the loss) and have lifetimes of less than 14 years (Ko et al., 2013). An additional criterion was that the HFCs have atmospheric records starting in 2005 or earlier. Thus, the HFCs chosen are: HFC-134a, HFC-152a, HFC-365mfc, HFC-245fa and HFC-32 (Table 1). These HFCs are used in a number of applications, specifically in refrigeration (HFC-134a, HFC-152a), air conditioning (HFC-134a, HFC-245fa, HFC-32), foam expansion (HFC-152a, HFC-245fa, HFC-365mfc), and as aerosol propellants (HFC-152a).


Our approach optimizes the OH oxidative capacity to reconcile modelled and observed mole fractions of three HFCs simultaneously. In addition, prior emission estimates of the three HFCs are also optimized. This approach relies on the assumption that errors in the prior emissions of each species are largely independent, whereas the effect of errors in the prior OH estimate on the HFCs is fully correlated. We use the AGAGE 12-box model (Cunnold et al., 2002; Rigby et al. 2013) to

model mole fractions of HFCs, and its adjoint model, with a Quasi-Newton algorithm (Thompson et al., 2018) to optimize OH and HFC emissions for the period 2000 to 2021. We chose this simple 12-box model to test the approach since it has a fast computation time (a few minutes for an inversion of 22 years), which allows multiple combinations of species to be tested and for ensembles to be run for uncertainty analysis. As a test for the robustness of our method, we examine the results using the ten possible combinations of three HFCs from our five selected species.

**2 Methodology**

**2.1 Modelling and inversion framework**

We used the AGAGE 12-box model of the atmosphere, and its adjoint, which has four equal area boxes in the latitudinal direction and three vertical layers representing the lower and upper troposphere and stratosphere (Cunnold et al., 2002; Thompson et al., 2018). We modified this model to simulate three HFCs and to optimize their emissions and the OH oxidative

capacity simultaneously. HFC mole fractions were modelled at monthly resolution using monthly resolved prior emissions and OH concentrations and species specific OH rate constants (Table S1). The emissions were optimized monthly for each of the four latitudinal boxes, and OH was optimized monthly for each of the twelve boxes. The optimal emissions and OH are found by minimizing the cost function (Tarantola, 2005):



$$J(x) = \frac{1}{2}(x - x_b)^T B^{-1}(x - x_b) + \frac{1}{2}(H(x) - y)^T R^{-1}(H(x) - y) \qquad (2)$$

where $x$ and $x_b$ are vectors of the posterior and prior emissions and OH oxidative capacity, respectively, $y$ is a vector of observed mole fractions for the three HFC species, $H(x)$ is the function of atmospheric transport, and B and R are the prior and observation error covariance matrices, respectively. The solution, $x$, to the cost function was found using the Quasi-Newton algorithm, M1QN3.

Initial mole fractions in each box for HFC-134a and HFC-152a were calculated by running the forward model for 22 years with constant emissions equalling the estimated global annual atmospheric loss to match the initial mole fractions measured in 2000. For the other species it was assumed that the atmospheric mole fraction was zero in the year 2000.

### 2.2 Atmospheric observations and data processing

We use observations from the Global Monitoring Laboratory of the National Oceanic and Atmospheric Administration (NOAA) network (https://gml.noaa.gov/aftp/data/), from the Advanced Global Atmospheric Gases Experiment (AGAGE) network (https://agage2.eas.gatech.edu/data_archive/) and, for HFC-245fa and HFC-365mfc, from the Cape Grim Air Archive (southern hemisphere, SH) and the Scripps Institution of Oceanography (SIO) collection of archived air samples (northern hemisphere, NH) (Vollmer et al., 2011) (Tables S2 and S3). NOAA observations are from paired flask samples taken at 12 relatively remote sites, which have been analysed using gas chromatography with mass spectrometry (GC-MS) and are available at weekly to biweekly frequency (Montzka et al., 2015). The NOAA data were averaged to monthly means. AGAGE observations are made in-situ using GC-MS instruments and are thus available at much higher temporal frequency (Prinn et al., 2018; https://agage.mit.edu). In this study, we have used the monthly mean data provided by AGAGE (which exclude observations influenced by local emissions). Initial AGAGE measurements of HFC-134a, HFC-152a, and HFC-365mfc at Mace Head (MHD), Cape Grim (CGO), Jungfraujoch (JFJ), and Zeppelin (ZEP) were performed with AGAGE ADS GC-MS systems (Vollmer et al., 2011; Simmonds et al., 1995). Starting from the end of 2004, new generation Medusa GC-MS systems were phased-in (Miller et al., 2008). Measurements at Monte Cimone were performed with a different GC-MS system (Maione et al., 2013). The archive data are from air samples taken from 1973 to 2010 at a small number of sites and were analysed on Medusa GC-MS systems at CSIRO and SIO (Vollmer et al., 2011).

Monthly means for each of the four latitudinal boxes were derived by averaging monthly data from all sites within each box. Outliers at each site (defined as outside ±2 standard deviations of the mean over a running time window of 12 months) were removed. The archive data at Cape Grim and Trinidad Head compared well with the AGAGE data for HFC-245fa and HFC-365mfc at these sites during overlapping years. Also, the AGAGE and NOAA data compared very well at the four sites where both networks' data are available, except for HFC-32 and HFC-365mfc. For HFC-32 and HFC-365mfc, the difference between the networks was significant and increased with increasing atmospheric abundance (the difference relative to abundance



showed no significant trend over 2000-2021). Therefore, we calculated a monthly dependent adjustment factor for HFC-32 and HFC-365mfc to adjust the NOAA observations to match those of AGAGE (Figure S1). This adjustment was calculated as a quadratic fit to the difference between the NOAA and AGAGE observations at sites where both networks' data are available (i.e., 4 sites). We then applied the adjustment across the 4 sites to correct the data at all sites. Since it is not known which of

the two scales is more accurate, we also include inversions in which the AGAGE data for HFC-32 and HFC-365mfc are adjusted to match those of NOAA (i.e. using the same quadratic fit). (The assimilated observation records for each species and box are shown in Figure S2).

The observation uncertainties used in the inversion need to represent the uncertainty of the monthly mean values for each box, that is, how well they represent the true means. Uncertainties were thus calculated for each month and box as the greater of

the standard deviation of mole fraction from all sites in that box or 1% of the mean monthly mean mole fraction. (The choice of 1% is based on the approximate measurement uncertainty). If this value was less than 0.5% of the global mean mole fraction for that month, then the latter was set as the minimum.

## 2.3 Emissions and other input data

Prior emissions for the four surface/lower troposphere boxes were taken from estimates from the Emission Database for Global Atmospheric Research (EDGARv7) (EDGAR, 2022). These were provided at annual temporal and at 0.1° spatial resolution and were averaged to each box and interpolated in time to monthly resolution. For the NH, prior uncertainties on the emissions were set to the greater of 50% of the value in each month and box or 50% of the mean over the whole timeseries for that box. This was done to increase the uncertainty early in the timeseries when the absolute emissions are small. For the SH, since the

emissions are very small, the uncertainties were set to the greater of 25% of the global mean value in each month or 25% of the global mean over the whole timeseries. This resulted in a mean global uncertainty of approximately 60% for all species. The emission errors in each box were assumed to be uncorrelated, however, we assumed a temporal correlation scale length of 60 months. This long correlation length was chosen as the year-to-year variations in the emissions should be relatively small and the emissions should evolve according to economic demand and policy surrounding their production and use. We also

performed sensitivity tests for this choice, which are described in Section 2.4.

Prior OH concentrations were provided monthly for each box based on the Spivakovsky et al. (2000) climatology and the temperature for each month and box was taken provided by the European Centre for Medium Range Weather Forecast (ECMWF) reanalysis data. The uncertainty on the prior OH concentrations were set to 60% in all boxes, which is equal to the

global mean uncertainty on the prior emissions.





## 2.4 Sensitivity tests

To test the sensitivity of the choice of uncertainties on the prior emissions and the OH concentration, we carried-out inversions with varying percentage uncertainties using the species combination HFC-134a, HFC-152a, and HFC-365mfc. These tests were: i) Using smaller but still balanced prior emission and OH uncertainties; prior emission uncertainties were calculated as described above but using 25% in the NH and 12.5% of the global mean emission in the SH, and the OH uncertainty was 30% (i.e., equal to the global mean emission uncertainty). ii) Using the same prior emission uncertainties as described above (50% in the NH and 25% of the global mean in the SH) but with a smaller OH uncertainty of 50%. iii) Using the same prior emissions uncertainties as in (ii), and OH uncertainty of 60% but with a temporal correlation scale length of 24 months. iv) As in (iii) but using a correlation scale length of 36 months. The posterior emissions were found to be insensitive to the choice of uncertainty (Figure S3). However, the amplitude of the variability in OH was sensitive to the OH uncertainty, with smaller amplitude using 50% uncertainty compared to 60%, and to the temporal correlation scale length of the fluxes, with smaller amplitude with scale lengths of 24 and 36 months. The temporal pattern of OH variability was unaffected though. We chose to use globally balanced OH and emission uncertainties at 60% and a temporal correlation length of 60 months, with the latter chosen to minimise small high-frequency variations in the posterior emissions of some species, which we consider to be noise.

We also carried-out a synthetic data test to determine how well the inversion can constrain OH. For this test, we generated timeseries of HFC-134a, HFC-152a, and HFC-365mfc mole fractions for each lower troposphere box and month using the EDGARv7 emissions and OH (generated by randomly perturbing the Spivakovsky OH estimates with temporal correlation length of 36 months). Random errors consistent with our observation uncertainties were added to the time series of HFC-134a, HFC-152a, and HFC-365mfc to form the synthetic observations. For the inversion, we used prior emissions generated by adding random errors (with error characteristics consistent with our prior emission uncertainties) to the EDGARv7 emissions. For the prior OH, we used the Spivakovsky climatological estimate. (Note, in the synthetic test the true emissions are EDGARv7 and the true OH is the randomly perturbed OH). The inversion captured the true emissions very well with reductions in the RMSE for HFC-134a, HFC-152a and HFC-365mfc of 19.3 to 6.8, 6.9 to 1.3 and 0.22 to 0.10, respectively. The inversion also captures the annual mean variability in OH quite well with correlation coefficients ($R^2$ values) for the posterior versus the true OH globally and for the NH and SH of 0.44, 0.76 and 0.83, respectively (Figure S4). There is a lag time of 2-3 years before the posterior OH departs from the prior value, which is due to the fact that the abundance of HFC-365mfc is still very low during these years and hence the sensitivity of OH in our inversion to this species is then also low.

## 3 Results

### 3.1 Atmospheric trends of HFC species

All HFC species in our study increased in the atmosphere from 2000 to 2021, but with differing temporal patterns, which reflect the different evolution of emissions and the different lifetimes of each species (Figure 1). HFC-134a exhibited a small



positive trend in its growth rate, which had a mean of 4.6 ppt/y and a trend of 0.10 ppt/y$^2$ throughout the 22-year study period. HFC-152a increased quasi-steadily at a rate of 0.50 ppt/y until 2008, when the rate decreased to less than 25% of its initial value. HFC-365mfc increased quasi-steadily at a rate of 0.06 ppt/y until 2017, when its growth rate decreased by nearly 50%. The abundance of HFC-245fa was below detection limit until 2003, after which it increased steadily with a mean growth rate of 0.17 ppt/y. And HFC-32 exhibited a positive trend in growth rate with a mean of 1.7 ppt/y and a trend of 0.22 ppt/y$^2$.

The simulated mole fractions using the prior emissions had quite different errors with respect the observations of each HFC indicating that the errors in the emissions are largely uncorrelated across the species. The prior emission estimate of HFC-134a leads to an overestimate the atmospheric increase throughout the study period, while that of HFC-152a gives a good agreement with the atmospheric trend until 2010 after which it leads to an overestimate of the growth. For HFC-365mfc, the prior emission estimate leads to an underestimate of the atmospheric increase in the first 10 years resulting in an underestimate of the atmospheric abundance until 2020. Similarly, for HFC-245fa the prior estimate leads to an underestimate of the atmospheric increase in the first 10 years but then gives a good agreement with the atmospheric abundance between 2015 and 2020. For HFC-32, the prior emissions provide a good agreement with the observations until 2010, and subsequently overestimate and then underestimate the atmospheric abundance.

The mole fractions modelled using the posterior fluxes, on the other hand, match the observations well for all years in our study period (Figure 1 and Figure S5). The root-mean-square-error (RMSE) of the model-observation differences were greatly reduced using the posterior versus the prior fluxes. The reduction in the RMSE (normalized by the global mean mole fraction of each species) from prior to posterior was for HFC-134a, HFC-152a, HFC-365mfc, HFC-245fa and HFC-32, respectively: 21.7% to 0.6%, 15.6% to 3.5%, 20.3% to 2.4%, 10.5% to 2.5%, and 9.9% to 2.4%

## 3.2 Emission estimates

Figure 2 shows the prior and posterior emissions estimates for each species. The range of posterior estimates from all 10 inversions (using the HFC-365mfc and HFC-32 observations in which the NOAA data were adjusted to match the AGAGE scale) is small, generally within ±5% of the mean (Figures S6). Using the dataset in which the AGAGE data for HFC-365mfc and HFC-32 were adjusted to the NOAA scale resulted in nearly the same emissions for these species, except for HFC-32 for which the increase in emissions was smaller after 2016 and followed closely the prior emissions.

Although the primary aim of our study is not to analyse the posterior emissions of the five HFC species, there are a few noteworthy features. For three species (HFC-152a, HFC-365mfc and HFC-245fa) the inversions find an appreciable decrease (or for HFC-245fa a plateauing) in the emissions starting from July 2007 and continuing until 2009. We propose that this could be a result of the global financial crisis, which led to a significant reduction in international trade and investment, which culminated in 2008 (IMF, 2009). The three species that appear to have been affected are used as foam expanding agents and





aerosol propellants, whereas the two species that were not affected (i.e., HFC-134a and HFC-32) are primarily used in refrigeration and air-conditioning.

The inversions also find a decrease in the emission of HFC-365mfc and a plateauing in the emissions of HFC-134a following

230    the signing of the Kigali Amendment (KA) in October 2016. The KA to the Montreal Protocol is an international agreement to reduce the consumption and production of HFCs, which are powerful greenhouse gases. Under the KA, most industrialized countries commit to a reduction in the production and consumption of HFCs of 10% by 2019, 40% by 2024 and 80% by 2034, while most developing economies commit to reductions of 10% by 2029 and 80% by 2045. We propose that the emissions of these HFCs may have reduced (or in the case of HFC-134a, plateaued) in response to the KA and in anticipation of tighter

235    controls on their production and use.

### 3.3 Estimate of OH oxidative capacity

Figure 3 shows the annual mean OH oxidative capacity for the troposphere estimated from our inversions. Note that we only show the OH results from 2004 because there are very few observations of HFC-365mfc and HFC-245fa and no observations of HFC-32 prior to 2005, and thus the OH estimated from the inversions remained close to the prior estimate for the first 4

240    years. We show the mean and range of results obtained with two sets of 10 inversions: i) using data in which HFC-365mfc and HFC-32 observations from NOAA were adjusted to match the AGAGE scale ("adjusted NOAA"), and ii) using data in which the AGAGE observations were adjusted to match the NOAA scale ("adjusted AGAGE"). We also show a third timeseries, namely, a subset selecting only the inversions that included the shorter-lived species, HFC-32 and HFC-152a. (In Figure S7, we show the OH estimate from individual inversions).

245

Up to 2017, all three timeseries are quite consistent. However, from 2017 the inversions using "adjusted NOAA" result in somewhat lower OH values compared to "adjusted AGAGE". This difference is largely due to differences in the HFC-32 observations between the two datasets, with the "adjusted NOAA" showing a stronger increase after 2017, which is matched in the model by increasing the emissions of HFC-32 but also by decreasing OH. The difference between these two timeseries

250    indicates the uncertainty in our OH estimate due to the scale uncertainties in HFC-32 and HFC-365mfc. The inversions including both HFC-32 and HFC-152a result in lower OH values from 2017 compared to the mean of all inversions (regardless of the dataset used). This may reflect the increased sensitivity of derived OH in our analysis to these shorter-lived species.

### 4 Discussion

### 4.1 Comparison with other OH estimates

255    We compared our OH estimate from inversions including both HFC-32 and HFC-152a to other estimates from ACTMs and from optimizations using MCF (Figure 4a). For the comparison, we selected studies that cover at least the period 2005-2015.



Overall, we find a smaller degree of variability in our annual OH anomaly (<2%) compared to the estimates shown based on MCF observations (<6%) (e.g. Patra et al., 2021; Naus et al., 2021), but consistent with the degree of variability found by Montzka et al. (2011) also using MCF. The magnitude of our variability is close to that estimated by Nicely et al. (2018) (<2%) using an ACTM and that from the Copernicus Atmospheric Monitoring Service (CAMS) reanalysis product, EAC4 (Inness et al., 2019) (<2.5%). The estimate of He et al. (2021) is also based on an ACTM but showed higher variability (<4%) than the other ACTM estimates.

EAC4 is the only estimate in Figure 4a that indicates an increasing trend, but this result is consistent with other ACTM estimates (not shown in Figure 4a since they end earlier than 2015) that indicate an increase in OH from at least 2005 (Dalsøren et al., 2016; Stevenson et al., 2020; Zhao et al., 2020). The other estimates, including our own, do not show any significant trend from 2005-2021.

Previous studies have found a link between OH variability and the El Niño Southern Oscillation (ENSO) with lower OH values during El Niño versus La Niña periods (Anderson et al., 2021; Patra et al., 2021; Zhao et al., 2020; Prinn et al., 2005). Although the mechanism for the correlation is not fully understood it is thought to be driven by changes in $NO_2$ and CO abundances (Anderson et al., 2021). Our OH estimate is weakly negatively correlated with the Multivariate ENSO Index (MEI), with lags of 6 and 9 months the correlation is r = -0.24 (p-value = $9\times10^{-4}$) and r = -0.33 (p-value = $6\times10^{-6}$), respectively (Figure 4b).

In our OH estimate from inversions including the two shortest lived species in our study (i.e., HFC-32 and HFC-152a), the year 2020 had the lowest OH value, with an anomaly of -1.6 ± 0.9% (where the uncertainty is the standard deviation of inversion results), which was lower by 0.6 ± 0.9% than that in 2019, but not significantly different from that derived for 2018. One other study using an ACTM has also pointed to a lower OH value in 2020 as a result of decreased NOx emissions during Covid-19 economic restrictions (Peng et al., 2022). However, our result for 2020 has a considerable uncertainty as the average anomaly of all inversions is only -0.5% ± 1.1% in 2020 and this year has a similar OH value to that in 2019.

## 4.2 Impact of OH on the CH$_4$ budget

Since CH$_4$ has been increasing in the atmosphere since 2007, with the highest ever measured annual increases in 2020, 2021 and 2022 (https://gml.noaa.gov/ccgg/trends_ch4/), there is considerable interest and urgency in better understanding changes in the OH oxidative capacity. Already, one study attributes some of the CH$_4$ increase in 2020 to a decline in OH concentration (Peng et al., 2022), while at present no study of 2021 is yet available. Other studies have proposed that a decline in OH from 2005 may be responsible for the general increasing trend in CH$_4$ in the atmosphere from 2007 (Rigby et al., 2017; Turner et al., 2019) but the magnitude of uncertainty in these studies is large and prevent a definitive conclusion on any trends.



We examine the impact of OH on the $CH_4$ budget using our estimate versus the climatology of Spivakovsky et al. (2000). For this, we again use the AGAGE 12-box model using prior $CH_4$ emissions as described in Table 2. For the spin-up we use the same procedure as for HFCs (see Section 2.1). For the inversion, we used observations at 42 sites in the NOAA discrete sampling network (https://gml.noaa.gov/aftp/data/), which were selected on the basis that they had observations quasi-continuously throughout the period 2005-2021 (see Table S5 for the list of sites used in each box). The observations were averaged to monthly means for each latitudinal box (see Figure S8). The prior emission uncertainty was set to 25% of the emission in each box and month with a temporal correlation scale length of 3 months.

Overall, the impact on the posterior emission estimates from using the climatological versus our optimized OH estimate is small (Figure 5). This is not unexpected since our OH estimate has only small interannual variations (<2%) and has no trend. There are, however, a few differences. Notably, using our OH estimate results in a smaller increase in emissions between 2019 and 2020 ($14 \pm 16$ Tg/y) compared to using the climatology ($23 \pm 16$ Tg/y). This is due to the decrease in our OH estimate in 2020 compared to 2019 (of 0.6%). In fact, the total $CH_4$ loss in 2020 is greater than in 2019 using both OH estimates, simply because the atmospheric $CH_4$ burden is greater in 2020, but our OH estimate has a 0.6% lower loss rate in 2020 versus 2019. In contrast, the large increase in atmospheric $CH_4$ in 2021 is not due to an anomaly in the loss rate, but to an even larger increase in emissions, $16 \pm 17$ Tg/y more than in 2020 according to our estimate.

## 5 Conclusions

Estimates of the atmospheric oxidative capacity due to OH have hitherto relied primarily on methylchloroform (MCF). However, as the atmospheric abundance of MCF has decreased the uncertainty in the OH derived with this approach has increased, meaning that alternatives to MCF will soon be needed. In this study, we have explored the potential to use hydrofluorocarbons (HFCs) in multi-species inversions to constrain OH. The HFCs, HFC-134a, HFC-152a, HFC-365mfc, HFC-245fa and HFC-32, were selected on the basis that their loss in the atmosphere is >99% due to OH oxidation and their lifetimes are relatively short, the longest being 14 years. Multiple inversions were performed each using three HFCs simultaneously, which were chosen from the five species listed above.

We find that the OH estimated from our inversions was very similar from 2004 to 2017 (within ±2%) regardless of the combination of species. After 2017, we found some differences in the results depending on whether the species HFC-32 and HFC-365mfc were adjusted to the AGAGE or NOAA scales, with inversions using data adjusted to the AGAGE scale giving on average 0.3% lower OH from 2018 to 2021. Also, inversions that included both HFC-32 and HFC-152a – the shortest-lived species in our study – resulted in even lower OH values (by 1%) from 2018 to 2021 (relative to early years) likely because they are more sensitive to inter-annual variations in OH. The inversions including HFC-32 and HFC-152a also found a negative anomaly in OH in 2020, $1.6 \pm 0.9\%$ below the mean for 2004 to 2021 and $0.6 \pm 0.9\%$ lower than in 2019, but the average



anomaly of all inversions was only $0.5 \pm 1.1\%$ lower in 2020. Overall, the inversions resulted in relatively small variations in OH, i.e., <2% and no trend over the period constrained by our inversions.

Using the OH estimate from our study an inversion of $CH_4$ resulted in similar global emissions compared to using a climatological OH estimate. With the OH estimate based on inversions including HFC-32 and HFC-152a, however, the resulting increase in $CH_4$ emission from 2019 to 2020 was 9 Tg/y smaller than that using the climatology owing to the lower OH loss rate in 2020.

In summary, we find that multi-tracer inversions using HFC species looks to be a promising method for constraining the atmospheric OH oxidative capacity and future work exploring this approach using 3D atmospheric chemistry transport models could lead to important additional insights.

**Author contribution**

RLT designed the study, wrote the code, analyzed the results and wrote the manuscript. SAM, MKV, JM, SR, PK, RGP and DY provided the measurements of HFCs and contributed to the manuscript, JA, CL, SD, IV, HW, and RFW provided
measurements.

**Competing interest statement**

The authors declare that they have no conflict of interest.

**Acknowledgements**

R. Thompson received financial support from the ReGAME project funded by the Research Council of Norway (grant no.
325610). S. Montzka acknowledges the technical assistance of B. Hall, C. Siso, S. Clingan, K. Petersen, D. Mondeel, F. Moore, and station personnel involved with collecting flask-air samples at sites across the globe. Measurements at Jungfraujoch are supported by the Swiss National Programs HALCLIM and CLIMGAS-CH (Swiss Federal Office for the Environment, FOEN) and by the International Foundation High Altitude Research Stations Jungfraujoch and Gornergrat (HFSJG). The AGAGE Medusa GC-MS system development, calibrations and measurements at the Scripps Institution of Oceanography, La Jolla and
Trinidad Head, CA, USA; Mace Head, Ireland; Ragged Point, Barbados; Cape Matatula, American Samoa; and Kennaook/Cape Grim, Australia were supported by the NASA Upper Atmospheric Research Program in the United States with grants NNX07AE89G and NNX16AC98G and 80NSSC21K1369 to MIT and NNX07AF09G, NNX07AE87G, NNX16AC96G, NNX16AC97G and 80NSSC21K1210 and 80NSSC21K1201 to SIO. The Department for Energy Security & Industrial Strategy (BEIS) in the United Kingdom supported the University of Bristol for operations at Mace Head, Ireland
(contract 1028/06/2015) and through the NASA award to MIT with sub-award to University of Bristol for Mace Head and Barbados (80NSSC21K1369). The National Oceanic and Atmospheric Administration (NOAA) in the United States supported



the University of Bristol for operations at Ragged Point, Barbados (contract 1305M319CNRMJ0028) and operations at Cape Matatula, American Samoa. In Australia, the Kennaook/Cape Grim operations were supported by the Commonwealth Scientific and Industrial Research Organization (CSIRO), the Bureau of Meteorology (Australia), the Department of Climate

Change, Energy, the Environment and Water (Australia), and Refrigerant Reclaim Australia, and through the NASA award to MIT with sub-award to CSIRO for Cape Grim (80NSSC21K1369). Measurements at Jungfraujoch are supported by the Swiss National Programs HALCLIM and CLIMGAS-CH (Swiss Federal Office for the Environment, FOEN) and by the International Foundation High Altitude Research Stations Jungfraujoch and Gornergrat (HFSJG). NOAA measurements were supported in part through the NOAA Cooperative Agreement with CIRES (NA17OAR4320101).

**Data availability**

The HFC and $CH_4$ emissions from EDGAR-v7 are available from the EDGAR website: https://edgar.jrc.ec.europa.eu/dataset_ghg70 (date of last access 20-Jul-2023). Other $CH_4$ emissions were from: i) the Global Fire Emissions Database (GFED) available from the database https://www.geo.vu.nl/~gwerf/GFED/GFED4/ (date of last access 20-Jul-2023), ii) the land-biosphere model LPX-Bern available on request to F. Joos (joos@climate.unibe.ch), the ocean

estimate of Weber et al. 2019 available from https://doi.org/10.6084/m9.figshare.9034451.v1 (date of last access 20-Jul-2023), the geological estimate of Etiope et al. 2019 available from https://gml.noaa.gov/ccgg/arc/index.php?id=130 (date of last access 20-Jul-2023), and termites estimate from Inness et al. 2022 available from the Copernicus Atmosphere Data Store https://ads.atmosphere.copernicus.eu (date of last access 20-Jul-2023).

The atmospheric observations of HFCs used in this paper were from: i) the NOAA GML network available from https://gml.noaa.gov/aftp/data/trace_gases/ (date of last access 20-Jul-2023), ii) the AGAGE network available from https://agage2.eas.gatech.edu/data_archive/ (date of last access 20-Jul-2023) and iii) atmospheric archive measurements available from Vollmer et al. 2011 (https://doi.org/10.1029/2010jd015309, date of last access 20-Jul-2023). The atmospheric observations of $CH_4$ used in this paper were from the NOAA GML network available from

https://gml.noaa.gov/aftp/data/greenhouse_gases/ (date of last access 20-Jul-2023).

**Code availability**

The AGAGE 12-box model and adjoint code used in this study are available from Zenodo: https://zenodo.org/record/8172277

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



**Table 1: Atmospheric lifetimes for the five HFC species used in this study. The prior and posterior lifetimes are those calculated by the AGAGE 12-box model and are determined by the OH concentration and rate constants.**

| Species | Lifetime Reference (years) | Lifetime Prior (years) | Lifetime Post (years) |
|---|---|---|---|
| HFC-134a | 13.4[1] | 14.5 | 14.5 |
| HFC-152a | 1.5[1] | 1.6 | 1.6 |
| HFC-365mfc | 9.9[2] | 7.1 | 7.2 |
| HFC-245fa | 7.7[1] | 7.5 | 7.5 |
| HFC-32 | 5.2[1] | 5.6 | 5.4 |

[1](Ko et al., 2013)
[2](Ehhalt et al., 2001)

**Table 2: Prior CH₄ emissions used in the AGAGE 12-box model. The totals are the mean for the period 2005 to 2021.**

| Source | Dataset | Temporal resolution | Total (Tg/y) |
|---|---|---|---|
| Anthropogenic | EDGAR-v7 (EDGAR, 2022) | Annual | 347 |
| Wetlands (net incl. mineral soil sink) | LPX-Bern[1] (Spahni et al., 2011) | Monthly | 206 |
| Biomass burning | GFED-v4.1s (Werf et al., 2017) | Monthly | 15 |
| Ocean | (Weber et al., 2019) | Monthly, climatology | 10 |
| Geological (land only) | (Etiope et al., 2019) | Annual, climatology | 15 |
| Termites | CAMS-GLOB-TERM-v1.1 (Doubalova and Sindelarova, 2018) | Monthly, climatology | 20 |
| Total | | | 613 |

[1]Since this estimate is only available until 2020, we used the 2020 estimate also for 2021.



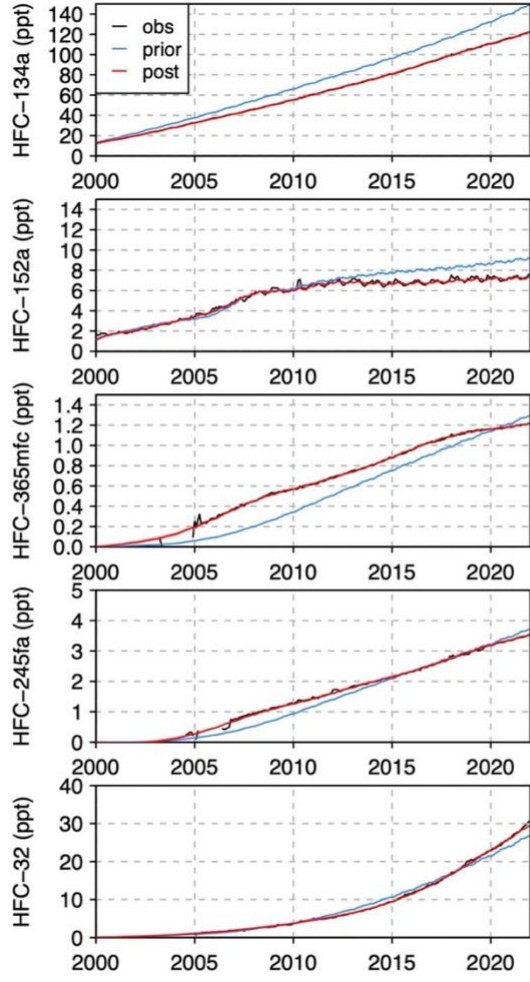

**Figure 1:** Time series of the modelled and observed global monthly mean mole fractions in the lower troposphere for the five HFC species used in this study. For HFC-32 and HFC-365mfc, the observations and posterior results are shown for the case in which the NOAA observations were adjusted to the AGAGE scale. The observations are shown in black and the modelled values using prior emissions are shown in blue. The posterior modelled values are shown in red and are the means of the six inversions that included the respective species and the ranges are indicated by the red shading (since the range for each species is very small the shading is hidden by the mean curves).





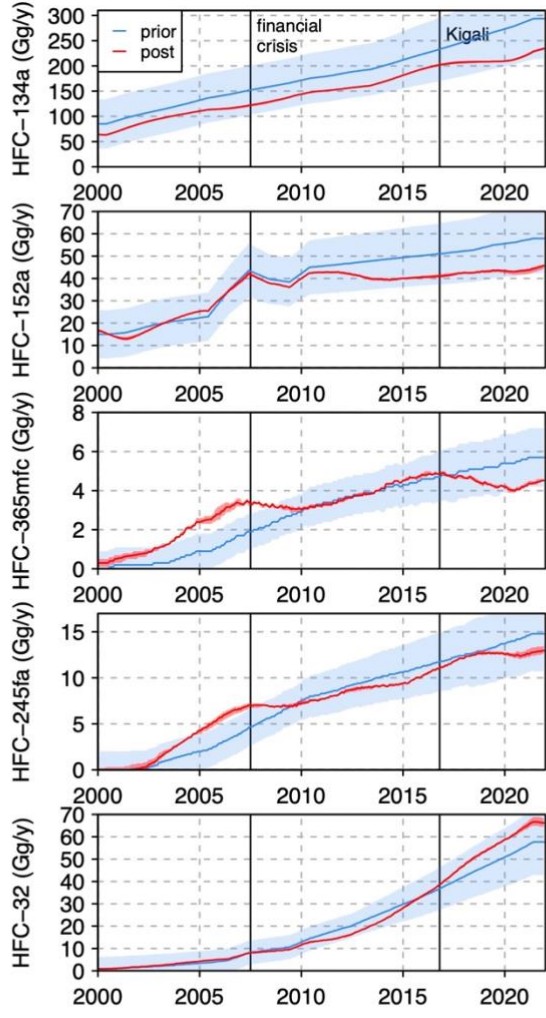

**Figure 2.** Time series of the global monthly mean emissions (Gg/y) for the five HFC species. Note the posterior results are shown for the case in which the NOAA observations of HFC-32 and HFC-365mfc were adjusted to the AGAGE scale. The posterior (red) curves are the means of the six inversions that included the respective species and ranges are indicated by the red shading (note that the range is very small and the shaded area is partially obscured by the mean curve). World events that may have influenced the emissions of some species are indicated by the vertical black lines: the start of the financial crisis of 2007-2008 and the signing of the Kigali Amendment to the Montreal Protocol in October 2016.






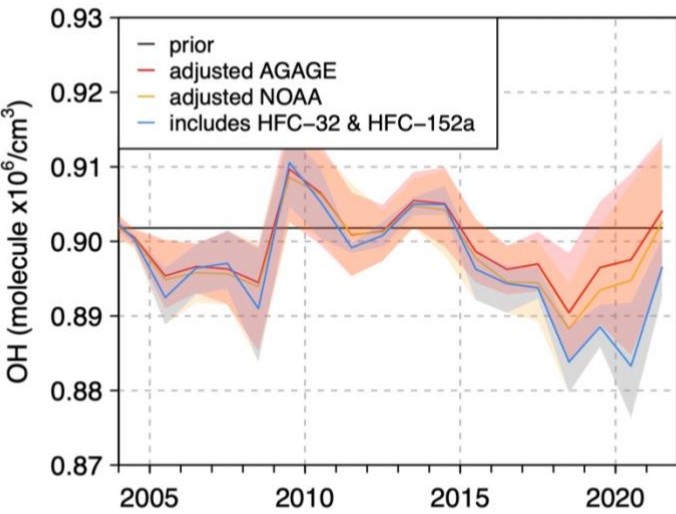

**Figure 3.** Time series of the global annual mean OH concentration shown for 2004-2021 (the period prior to 2004 is not shown as the posterior estimates remain close to the prior owing to the time lag in the sensitivity of OH and because there are very few observations of HFC-365mfc and HFC-245fa and no observations of HFC-32 prior to 2005. The mean estimates for inversions using two different sets of observations for HFC-32 and HFC-365mfc are shown: i) adjustments made to the AGAGE data to match the NOAA calibration scale (red) 565 and ii) adjustments made to NOAA data to match the AGAGE calibration scale (orange), as well as the mean of inversions that include both the shorter-lived species, HFC-32 and HFC-152a (blue). The prior OH is shown by the black line.





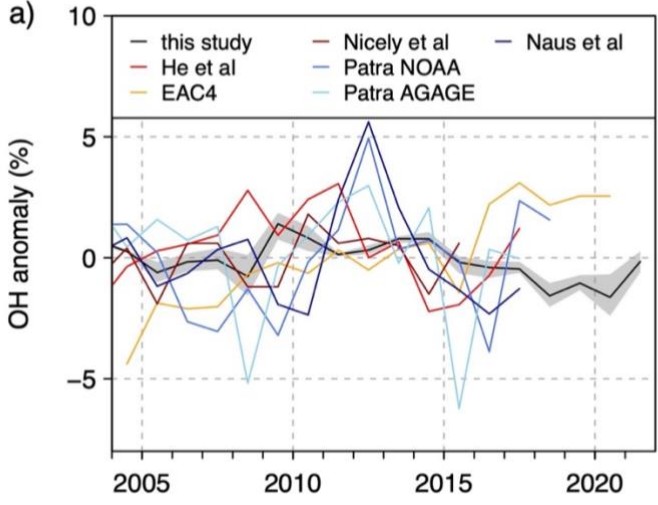


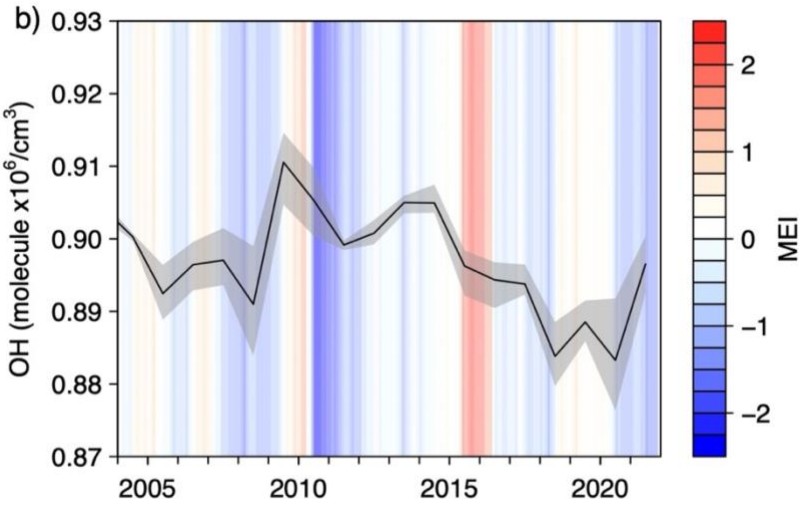

**Figure 4.** Timeseries of the global annual mean OH concentration from this study (mean of all inversions including HFC-32 and HFC-152a). Comparison with independent estimates from ACTMs (He et al., 2021; Nicely et al., 2018) and EAC4 ((Inness et al., 2019) and based on optimizations using MCF (Naus et al., 2021) using NOAA observations; and the two estimates from (Patra et al., 2021) using NOAA and AGAGE observations separately) (a). Comparison of the estimate from this study with the monthly multivariate ENSO index (MEI) with red indicating El Niño and blue La Niña phases (b).




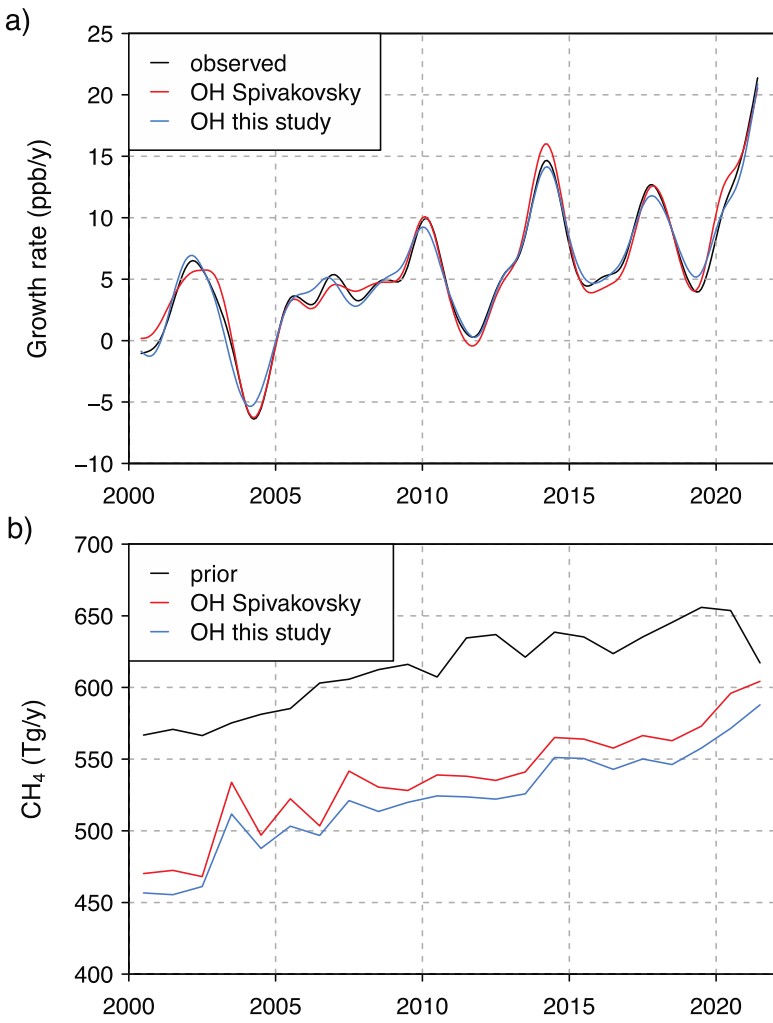


**Figure 5.** Global annual CH$_4$ growth rate from observations and using the AGAGE model using the climatological versus our estimate of OH (a) and the global source estimated from inversions using climatological versus our estimate of OH (b).