# Peer review of "Estimation of the atmospheric hydroxyl radical oxidative capacity using multiple hydrofluorocarbons (HFCs)"

_EGUsphere, 2023_

## Author Comment (AC1)

**Reply to Reviewer 2**

*Please note that the reviewer's comments are included in normal font and our responses are in italics below.*

General comments

The paper describes the setup and result of an inverse modelling study to derive global hyrdoxyl radical (OH) concentrations from hydrofluorcarbon (HFC) observations for the period 2004-2021. Concentrations of OH are important for the estimation of the global CH4 budget, which is the second most important greenhouse gas after CO2, and as OH is one of the main uncertainties in there, this study is of great relevance. The method is based on the idea that emissions of selected HFCs are more or less known, and that destruction by OH is their only loss-process; observations of these HFCs are then used to decrease the uncertainty in OH concentrations as well as emissions. Similar approaches have been used in combination with methylcloroform (MCF) and to some extend also with HFCs, but this study is unique it its focus on 5 different HFCs and the long time span of 2004-2021.

A strong element of this study is its completeness. Inversion experiments have been performed for all 10 possible combinations of 3 out of 5 available HFCs observation time series, 2 adjustments of observation scales have been compared, and also sensitivity experiments were done. The resulting range of estimated OH concentrations are compared with results obtained in other studies, and found to be in good agreement. Special attention is also paid the possible impact of the 2021 lockdowns on concentrations. In addition, also the impact of the result on CH4 concentrations is discussed.

*We thank the reviewer for this positive assessment of our study and reply to the question/comments below.*

The study uses a 12-box model to simulate global atmospheric mixing. Although this is a very crude model, it makes sense here since the amount of observations is limited, and results are discussed at global scale only. Could the authors indicate what is the expected gain from using a full CTM, as suggested in the discussion at line 330?

*The shortest-lived species in our study has a lifetime of 1.5 years (HFC-152a), thus its variability at hemispheric scale should be able to be represented as the mean of a few sites as well as by our model. However, using a 3D ACTM would enable individual sites to be represented and the differences between sites may provide more information on the variability of OH within a semi-hemisphere, which our model is not able to do. Also a 3D ACTM driven by reanalysis meteorology should be able to capture the influence of climate oscillations, e.g. ENSO, on atmospheric transport. We have now included this explanation at the end of the Conclusions section.*

The current paper does not discuss any results at the level of the 12-boxes, for example by latitude band or by atmospheric layers. Could useful information already be found in here, for example on the difference between tropo- and stratosphere?

[Figure]

*Figure 1. Prior (blue) and posterior (red) OH concentrations from inversions including HFC-32 and HFC-152a shown for the northern hemisphere (top) and southern hemisphere (bottom).*

*There is virtually no sensitivity of our observations to OH in the stratosphere, thus for the uppermost layer (stratosphere) the posterior OH remains very close to the prior values. In the upper troposphere (the middle layer) there is a weak sensitivity, and the posterior OH changes a little compared to the prior, but most of the change posterior versus prior is in the lower troposphere.*

*We analysed the posterior OH for the troposphere by hemisphere but decided not to include these results in the paper, but rather to focus on the variations in global OH. The posterior OH in each hemisphere has the same degree of variability as the global mean, i.e., <2%, and the variations between 2009 and 2015 are positively correlated in both hemispheres (Fig.1 above). For the inversions including the shorter-lived species, HFC-152a and HFC-32 (same inversions as shown by the blue curve in Fig. 3 in the manuscript), the decrease in global OH in 2018 is mostly driven by the NH, but the decrease in 2020 is entirely driven by the SH (see Fig. 1 above). In 2020 there was a reduction in NOx emissions globally, which could also act to reduce OH, except in very polluted areas (Zhu et al., 2022). The biggest reductions in NOx emissions were in the NH, in particular, over Europe and East Asia (especially around Beijing) (Cooper et al. 2022). However, considering Beijing is a very polluted area, and both this region and Europe are outside the region of the highest OH concentrations, it is unclear how strong the impact of this would be on global OH. At the same time, there was also a reduction in NOx in the tropics and SH, which corresponds to an observed reduction in biomass burning in 2020, thus it is possible that there could have been an NOx induced reduction in OH in the SH, but we have fairly low confidence in this result.*

*References:*

*Zhu et al. Estimate of OH trends over one decade in North American cities. PNAS, **119**(16), (2022). https://doi.org/10.1073/pnas.2117399119*

*Cooper et al. Global fine-scale changes in ambient $NO_2$ during COVID-19 lockdowns. Nature, **601**, 380–387 (2022). https://doi.org/10.1038/s41586-021-04229-0*

For the discussion it might be useful to spend some lines on the long term perspective of OH studies. A major driver for this study was the current amount of MCF remaining in the atmosphere these days is too small to be used as proxy for OH. Due to the Kigali Amendement to the Montreal Protocol, also emissions of HFCs are expected to decrease and eventually disappear. What is the time range for which the authors expect that their method can be used? And do they have suggestions for observations that should be soon setup to ensure that also in following decades estimates of the OH concentrations can still be made?

*Based on the recent HFC scenarios of Velders et al. (2022), emissions of HFCs are expected to continue up to at least 2050, although at a lower level compared to without the Kigali Amendment. Thus, we think the method could be still used up to the middle of this century at least. We have included the following text in the Conclusions section, which we have renamed "Conclusions and outlook":*

*"One important question though, is how the atmospheric mixing ratios of these HFC species will evolve in the future considering the Kigali Amendment (KA) to the Montreal Protocol. The KA is expected to reduce the $CO_2$-equivalent emission of HFCs in 2050 to ~1 $GtCO_2$ eq $yr^{-1}$, that is, by 30 to 50% compared to that expected in the absence of the KA, assuming that the KA is adopted by all countries (Velders et al., 2022), with HFCs with large Global Warming Potentials contributing more to the reduction. Thus, emissions of HFCs will likely continue up to at least 2050, although at a reduced level, meaning that the atmospheric mixing ratios should start to stabilize, or in the case of HFC-152a, start to go down. As such, this method should be applicable up to the middle of this century."*

*Concerning the second part of this question "suggestions for observations that should be soon setup", we currently do not have any suggestions for new measurements to set-up to better constrain OH.*

*Reference:*
*Velders, G. J. M., et al., Atmos Chem Phys, 22, 6087–6101, (2022). https://doi.org/10.5194/acp-22-6087-2022.*

Specific comments

line 101: Could some main characteristics of of H(x), the AGAGE 12-box model, be described here? For example, how is the exchange between the boxes parameterized?

*We have added the following description:*

*"Atmospheric transport was calculated as horizontal and vertical mixing rates between the boxes, which were provided monthly based on Cunnold et al. (2002) and the model was integrated in 2-day time steps using the fourth-order vectorized Runge-Kutta algorithm. Chemical loss, which in this study is solely due to OH, was calculated for each box and time step."*

line 152-153: The temporal correlation scale length that is described here, is that in combination  with an e-folding shape?

*The temporal correlation was calculated as an exponential decay with time with a scale length of 60 months. We have added this specification to the manuscript.*

line 159: The "reanalysis data" mentioned here, is that ERA5?

*Yes, it was ERA5, we have added this specification.*

line 159: There are no correlations assumed in the prior OH concentration uncertainties?

*That is correct, we assumed no correlation in the prior OH concentration uncertainties.*

line 274: correlation with ENSO Index in Figure 4b: maybe a scatter plot (with connected dots?) would be more clear here to show the (lack of) correlation.

*We prepared the following scatter plots in which we used monthly values of MEI and OH anomaly (with seasonality subtracted and smoothed with a spline fit to reduce the noise). Since the correlation is weak, we do not include the scatter plots in the manuscript but only in the supplement.*

[Figure]

Spelling and grammar

It was a pleasure reading the paper, it is very well written and illustrated. Textual comments are therefore very limited.

line 152: "The emission errors in each box were assumed to be uncorrelated *with other boxes*, ...

*We have added "with other boxes"*

line 157: ".. the temperature for each month and box was taken *from* the European Centre ..."

*We have corrected this.*

---

## Author Comment (AC2)

**Reply to Reviewer 1**

*Please note that the reviewer's comments are included in normal font and our responses are in italics below.*

General Comments

This manuscript presents an inversion study of five HFCs (different combinations of three at a time) to infer global annual mean hydroxyl radical (OH) concentrations using a 12-box model. The inferred OH anomalies are compared against other estimates from past MCF and CTM-based analyses. Finally, the impact of the optimized OH on the growth rate and emissions of $CH_4$ is derived from the same 12-box inversion relative to a Spivakovsky et al. climatology. The results suggest that variability in the annual OH anomaly is less than 2% with no trend over the period 2004-2021, that OH abundance in year 2020 was likely low but not significantly lower than in prior years (2018 especially), and that $CH_4$ emissions using the optimized OH had a smaller increase than is inferred using climatology, though the difference is small.

Overall, this is a compelling study focused on an important topic. The global oxidizing capacity is a subject of much debate, and further observational constraints to quantify it are always needed. The methodology used is sound, and I see no shortcomings in what is presented. I point out in my comments an opportunity for expanded discussion and a couple small clarifications, but otherwise, I think the article is well-presented, is of interest to readers of ACP, and represents a significant advance beyond the use of MCF as the main observational proxy of OH.

*We thank the reviewer for the positive feedback and reply to the specific comments below.*

Specific Comments

L39: Turner et al., PNAS, 2017 (https://doi.org/10.1073/pnas.1616020114) could be added to the list of MCF studies

*We agree and have added Turner et al. to the list of MCF studies.*

L275: It would be informative to expand on the discussion of the "shortest lived species" a bit. It is stated earlier in the manuscript that the derived OH may be more sensitive to the shorter-lived species in the inversion. Wouldn't it also make sense that, in the actual atmosphere, the shorter-lived species would adjust more quickly to either changes in emissions or variations in OH? This is not explicitly stated in the text, but since the authors separated out inversions that included the shorter-lived species (e.g., in Fig. 3), why not discuss the implications more?

*We agree that the discussion on the results using the shorter-lived species should be expanded considering the difference between the OH estimates using HFC-32 and HFC-152a (the shorter-lived species) and the other estimates. We have added some explanation to the last paragraph of section 3.3, where this difference in results is first mentioned. It is true that*

*the concentration of a shorter-lived species will have a stronger response to a change in OH. However, this species will be less sensitive to a change in emissions due to the greater loss rate.*

Figure S1: For the lower row of panels in each set, I think it would help to indicate on the y-axis that this represents a difference (something like "HFC-32 Difference, AGAGE – NOAA (ppt)" or similar).

*We have added "Δ" to the y-axis labels to indicate that this is a difference.*

Technical Corrections

Figure S5: Should the caption state "ten inversions" rather than six?

*No, the caption is correct, although there are 10 inversions in total, each species is included in only a sub-set of 6 inversions. There are 10 possible ways of selecting 3 species from a set of 5, so some species are excluded in some inversions.*

Figure S6: Figure seems low resolution, e.g. when compared to Figure S5.

*We think this is a rendering issue of the pdf file, which may be related to the fact that Fig.S5 contains transparency (i.e., the grey uncertainty envelope). The original jpg has the same resolution as the other figures.*